# Transdiagnostic Perspective of Impulsivity and Compulsivity in Obesity: From Cognitive Profile to Self-Reported Dimensions in Clinical Samples with and without Diabetes

**DOI:** 10.3390/nu13124426

**Published:** 2021-12-10

**Authors:** Giulia Testa, Bernat Mora-Maltas, Lucía Camacho-Barcia, Roser Granero, Ignacio Lucas, Zaida Agüera, Susana Jiménez-Murcia, Rosa Baños, Valerie Bertaina-Anglade, Cristina Botella, Mònica Bulló, Felipe F. Casanueva, Søren Dalsgaard, José-Manuel Fernández-Real, Barbara Franke, Gema Frühbeck, Montserrat Fitó, Carlos Gómez-Martínez, Xavier Pintó, Geert Poelmans, Francisco J. Tinahones, Rafael de la Torre, Jordi Salas-Salvadó, Lluis Serra-Majem, Stephanie Vos, Theresa Wimberley, Fernando Fernández-Aranda

**Affiliations:** 1Department of Psychiatry, University Hospital of Bellvitge, L’Hospitalet de Llobregat, 08907 Barcelona, Spain; gtesta@idibell.cat (G.T.); bmora@idibell.cat (B.M.-M.); lcamacho@idibell.cat (L.C.-B.); ilucas@idibell.cat (I.L.); zaguera@ub.edu (Z.A.); sjimenez@bellvitgehospital.cat (S.J.-M.); 2Psychiatry and Mental Health Group, Neuroscience Program, Institut d’Investigació Biomèdica de Bellvitge (IDIBELL), L’Hospitalet de Llobregat, 08907 Barcelona, Spain; 3Consorcio CIBER, M.P. Fisiopatología de la Obesidad y Nutrición (CIBERObn), Instituto de Salud Carlos III (ISCIII), 28029 Madrid, Spain; roser.granero@uab.cat (R.G.); rosa.banos@uv.es (R.B.); botella@psb.uji.es (C.B.); monica.bullo@urv.cat (M.B.); felipe.casanueva@usc.es (F.F.C.); jmfreal@idibgi.org (J.-M.F.-R.); gfruhbeck@unav.es (G.F.); mfito@imim.es (M.F.); carlos.gomez@urv.cat (C.G.-M.); xpinto@bellvitgehospital.cat (X.P.); fjtinahones@hotmail.com (F.J.T.); RTorre@imim.es (R.d.l.T.); jordi.salas@urv.cat (J.S.-S.); lluis.serra@ulpgc.es (L.S.-M.); 4Department of Psychobiology and Methodology, Autonomous University of Barcelona, 08193 Barcelona, Spain; 5Department of Public Health, Mental Health and Perinatal Nursing, School of Nursing, University of Barcelona, L’Hospitalet de Llobregat, 08907 Barcelona, Spain; 6Department of Clinical Sciences, School of Medicine and Health Sciences, University of Barcelona, L’Hospitalet de Llobregat, 08907 Barcelona, Spain; 7Instituto Polibienestar, Universitat de Valencia, 46010 Valencia, Spain; 8Core Lab Department, Biotrial Neurosciences, 35000 Rennes, France; valrie.bertaina-anglade@biotrial.com; 9Department of Basic Psychology Clinic and Psychobiology, Universitat Jaume I, Castellón de la Plana, 12071 Castellón, Spain; 10Department of Biochemistry and Biotechnology, Faculty of Medicine and Health Sciences, University Rovira i Virgili (URV), 43201 Reus, Spain; 11Institut d’Investigació Sanitaria Pere Virgili (IISPV), Hospital Universitari de Sant Joan de Reus, 43204 Reus, Spain; 12Molecular and Cellular Endocrinology Group, Instituto de Investigacion Sanitaria de Santiago de Compostela (IDIS), Complejo Hospitalario Universitario de Santiago de Compostela (CHUS), Santiago de Compostela University (USC) and Centro de Investigacion Biomedica en Red Fisiopatologia de la Obesidad Y Nutricion (Ciberobn), 15705 Santiago de Compostela A Coruña, Spain; 13National Centre for Register-Based Research, Department of Economics and Business Economics, Business and Social Sciences, Aarhus University and iPSYCH, The Lundbeck Foundation Initiative for Integrative Psychiatric Research (Copenhagen-Aarhus), DK-8210 Aarhus, Denmark; sdalsgaard@econ.au.dk; 14Department of Medical Sciences, School of Medicine, Hospital of Girona Dr. Josep Trueta, University of Girona, 17004 Girona, Spain; 15Departments of Human Genetics and Psychiatry, Donders Institute for Brain, Cognition and Behaviour, Radboud University Medical Center, 6525 GA Nijmegen, The Netherlands; Barbara.franke@radboudumc.nl; 16Department of Endocrinology, Instituto de Investigación Sanitaria de Navarra, University of Navarra (IdiSNA), 31008 Pamplona, Spain; 17Unit of Cardiovascular Risk and Nutrition, Hospital del Mar Institute for Medical Research (IMIM), 08003 Barcelona, Spain; 18Universitat Rovira i Virgili, Departament de Bioquímica i Biotecnologia, Unitat de Nutrició, 43201 Reus, Spain; 19Lipids and Vascular Risk Unit, Internal Medicine, University Hospital of Bellvitge (IDIBELL), L’Hospitalet de Llobregat, 08907 Barcelona, Spain; 20Department of Human Genetics, Radboud University Medical Center, 6525 GA Nijmegen, The Netherlands; geert.poelmans@radboudumc.nl; 21Department of Endocrinology and Nutrition, Virgen de la Victoria Hospital, Institute of Biomedical Research in Malaga (IBIMA), University of Malaga, 29016 Málaga, Spain; 22Integrative Pharmacology and Systems Neurosciences Research Group, Institut Hospital del Mar de Investigaciones Médicas Municipal d’Investigació Mèdica (IMIM), 08003 Barcelona, Spain; 23IMIM-Hospital del Mar Medical Research Institute and CIBER of Physiopathology of Obesity and Nutrition (CIBEROBN), University Pompeu Fabra (DCEXS-UPF), 08003 Barcelona, Spain; 24Nutrition Unit, University Hospital of Sant Joan de Reus, 43204 Reus, Spain; 25Nutrition Research Group, Research Institute of Biomedical and Health Sciences (IUIBS), University of Las Palmas de Gran Canaria, 35001 Las Palmas de Gran Canaria, Spain; 26Alzheimer Centrum Limburg, Department of Psychiatry and Neuropsychology, School for Mental Health and Neuroscience, Maastricht University, 6211 LK Maastricht, The Netherlands; s.vos@maastrichtuniversity.nl; 27National Centre for Register-Based Research, Department of Economics and Business Economics, Aarhus University, DK-8000 Aarhus, Denmark; tw@econ.au.dk

**Keywords:** impulsivity, compulsivity, decision making, cognitive flexibility, type 2 diabetes, novelty seeking, harm avoidance

## Abstract

Impulsive and compulsive behaviors have both been observed in individuals with obesity. The co-occurrence of obesity and type 2 diabetes (T2D) is more strongly associated with impulsivity, although there are no conclusive results yet. A multidimensional assessment of impulsivity and compulsivity was conducted in individuals with obesity in the absence or presence of T2D, compared with healthy, normal-weight individuals, with highly impulsive patients (gambling disorders), and with highly compulsive patients (anorexia nervosa). Decision making and novelty seeking were used to measure impulsivity, and cognitive flexibility and harm avoidance were used for compulsivity. For impulsivity, patients with obesity and T2D showed poorer decision-making ability compared with healthy individuals. For compulsivity, individuals with only obesity presented less cognitive flexibility and high harm avoidance; these dimensions were not associated with obesity with T2D. This study contributes to the knowledge of the mechanisms associated with diabetes and its association with impulsive–compulsive behaviors, confirming the hypothesis that patients with obesity and T2D would be characterized by higher levels of impulsivity.

## 1. Introduction

The prevalence of obesity worldwide has alarmingly increased, having nearly tripled in the last 50 years, reaching pandemic levels [1]. As one of the major risk factors for noncommunicable diseases, it has been associated with a reduced quality of life, high presence of disabilities, and decreased life expectancy [2]. Obesity is a complex disorder, usually classified as a metabolic, nutritional, and endocrine disease. Several factors contribute to the physiopathology of this disease, including genetic, social, environmental and psychological aspects [3]. According to the body mass index (BMI), obesity is classified in the following three categories: class I obesity (BMI: 30–34.9 kg/m^2^), class II obesity (BMI: 35–39.9 kg/m^2^), and class III obesity, or morbid obesity (BMI > 39.9 kg/m^2^) [4]. The presence of obesity is associated with multiple comorbidities that significantly contribute to higher rates of morbidity and mortality, including type 2 diabetes (T2D) and insulin resistance (IR), among others [5,6,7].

Excessive food consumption is one of the main contributors to weight gain in obesity. However, appetite and feeding behavior are not only controlled by energy requirements or metabolic need. Food also acts as a natural reinforcer, and its consumption is motivated by its hedonic properties, which rely on mesolimbic dopamine and opioids systems [8,9]. Processed foods, high in fats, sugars, and salt, are believed to stimulate appetite and increase calorie consumption through stimulation of opiates and dopamine receptors in the reward center [10,11]. Given the complexity and multicausality of this pathology, understanding the neurobehavioral mechanisms underpinning obesity is crucial to develop effective specific treatments.

Two constructs that have been suggested to play a role in excessive food intake and weight gain are impulsivity and compulsivity [12]. Impulsivity is typically defined as a tendency to act rashly without giving adequate forethought to the consequences of the behaviors, which, in the case of obesity, is reflected by overeating palatable foods [13]. Impulsivity is multidimensional, including personality traits (e.g., sensation seeking, lack of premeditation, and urgency) [14,15], motor impulsivity (e.g., response inhibition), and choice impulsivity (e.g., decision making and deficits in delay gratification) [16,17,18]. By contrast, compulsivity is characterized by repetitive and persistent behaviors, often harmful, despite their consequences [19]. In the context of overeating and obesity, this is reflected by repetition of maladaptive habits and a failure to shift behavior, despite its negative effects [20]. An important dimension of compulsivity is cognitive flexibility, which is the ability to flexibly adjust behavior to the demands of a changing environment (e.g., attentional set-shifting and task-shifting) [21,22].

Currently, there is a growing interest in analyzing dimensional models, where a spectrum around a specific construct will be considered, in which different disorders share some characteristics. From this point of view, the term dimension is understood as the set of magnitudes that serve to define a psychological phenomenon [23]. Thus, while the categorical model is based on the process of counting symptoms to an arbitrary number, where the presence of more symptoms becomes meaningless, in dimensional approaches, the number of diagnostic features forms an index of severity by taking into account the daily functioning of patients. The clinical utility of adopting dimensional models has been suggested, especially in the case of personality pathologies [24].

This is the case for the impulsive–compulsive spectrum, in which the dimensional approach is especially relevant. Along this spectrum, some mental disorders typically described in the impulsive pole are gambling disorder (GD) and other impulse control disorders, attention-deficit hyperactivity disorder (ADHD), borderline personality disorder, among others [25,26,27]. Compulsivity is well represented by anorexia nervosa restrictive type (AN-R), obsessive-compulsive disorder, and obsessive-compulsive personality trait [28,29]. Nonetheless, where obesity and obesity plus T2D comorbidity can be placed along the impulsive–compulsive spectrum is still unknown, which could have important implications for developing specific treatments.

Impulsive personality traits have been associated with a greater body mass index (BMI) and weight gain [30,31]. Moreover, strong evidence exists for a positive relation between obesity and cognitive indices of impulsivity, such as poor decision making [32] and deficits in delay gratification [33]. Similarly, a lack of cognitive flexibility has been shown in individuals with obesity and overweight [34,35]. Personality traits related to compulsivity, such as obsessive-compulsive traits and harm avoidance [36], as well as the ability to cope with negative emotions [37], have been suggested to play an important role in the development and perpetuation of obesity [38,39]. Accordingly, some studies showed elevated harm avoidance in individuals with obesity [40,41,42,43].

Type 2 diabetes is a metabolic disorder, characterized by pancreatic β-cell dysfunction and insulin resistance, which result in elevated levels of blood glucose [44]. Impaired glycemic control and IR have been suggested to impact brain dopaminergic systems [45,46,47,48,49,50], which may contribute to impulsivity and deficits in self-regulation, as well as impairment in cognitive functioning [51,52,53,54]. Although there are still no conclusive results, some studies highlight impairments in impulsivity, specifically in motor impulsivity in older adults with T2D [54], and recent research showed more disadvantageous decision making in T2D than in healthy controls in the Iowa gambling task (IGT) [55]. To the best of our knowledge, there are no current studies evaluating the association between compulsivity and T2D in individuals with obesity. Taking all this into account, it is unclear whether the presence of T2D affects different dimensions of impulsivity and compulsivity in individuals with obesity.

The present study aimed to describe and compare different clinical populations along the impulsivity–compulsivity spectrum. It especially focuses on individuals with obesity in the absence or presence of T2D, when compared with highly impulsive patients (namely, patients with GD) highly compulsive patients (namely, patients with AN-R), and healthy, normal-weight individuals. A multiple assessment of various impulsivity and compulsivity dimensions, using self-reported measures and neuropsychological tasks, was conducted to evaluate decision making and novelty seeking as markers of impulsivity, and cognitive flexibility and harm avoidance as markers of compulsivity. Based on the above-mentioned literature, individuals with obesity were expected to present compulsivity-related personality traits and poor cognitive flexibility. For the impulsivity dimensions, we hypothesize an impulsive profile to characterize obesity with T2D, with impulsive decision making and novelty seeking possibly being more pronounced in these individuals than in those with obesity only.

## 2. Materials and Methods

### 2.1. Study Design and Population

In the present cross-sectional study, a total of 581 participants along the impulsive–compulsive spectrum were included, as follows: *n* = 115 individuals with morbid obesity without diabetes (OB-DM), *n* = 67 individuals with morbid obesity and T2D (OB + DM), *n* = 107 individuals with anorexia nervosa restrictive subtype (AN-R), *n* = 121 individuals with gambling disorder (GD) and *n* = 171 healthy controls (HC). Participants in the AN-R and GD groups who presented with T2D were not included. Seven centers, all part of the Spanish Biomedical Research Centre in Physiopathology of Obesity and Nutrition (CIBERobn), participated in the study. The clinical groups were patients who had been consecutively referred to the clinics mentioned above. Healthy controls were recruited by means of word-of-mouth and advertisements at local universities, from the same catchment area as the clinical groups. The study was conducted according to the guidelines of the Declaration of Helsinki and its amendments, the International Conference on Harmonization Good Clinical Practice guidelines, and local regulatory requirements. The study was approved by the ethics committees of all participating institutions. Informed consent was obtained from all subjects participating in the study.

### 2.2. Psychometric Measures

The Temperament and Character Inventory—Revised (TCI-R) [56], previously validated in a Spanish adult population [57], consists of 240 items with a five-point Likert scale format. Three character dimensions are evaluated (self-directedness, cooperativeness, and self-transcendence) and also four temperaments (harm avoidance, novelty seeking, reward dependence, and persistence). In this study, harm avoidance and novelty seeking subscales were adopted as measures of compulsivity and impulsivity, respectively. For this sample, the Cronbach’s alpha was good, ranging from α = 0.830 (for novelty seeking) to α = 0.889 (for harm avoidance).

The Symptom Checklist—90 Items—Revised (SCL-90-R) [58], which was validated in a Spanish population [59], was administered for evaluating self-reported psychological distress and psychopathology. The instrument is scored on nine primary symptom dimensions (somatization, obsessive-compulsive behavior, interpersonal sensitivity, depression, anxiety, hostility, phobic anxiety, paranoid ideation, and psychoticism) and three global indices (global severity index (GSI), positive symptom total (PST), and positive symptom distress index (PSDI)). The internal consistency for the global index in our sample was high, α = 0.980.

### 2.3. Neuropsychological Measures

The computerized version of the Wisconsin card sorting test (WCST) [60] was used to evaluate cognitive flexibility through a set-shifting task. The WCST consists of matching stimulus cards within one of three of the following available categories: color, shape, or number. For a correct match, participants must identify the sorting rule, receiving the feedback of “right” or “wrong” after each sort. Following 10 consecutive correct matches, the rule is changed and then a new sorting rule must be identified. There are up to six attempts to detect the sorting rule and five rule shifts during the task. Each rule attainment is referred to as “category completed”. Participants do not know the correct rules or changes. The test continues until 128 cards are sorted. The following variables were adopted to measure cognitive flexibility: perseverative errors (i.e., failure to change sorting strategy after negative feedback), non-perseverative errors and the number of completed categories.

The Iowa gambling task (IGT) [61] is a computerized task proposed as a measure of choice impulsivity as it evaluates decision making. It is performed by selecting between four decks where each deck provides a specific amount of play money. It consists of a total of 100 turns in which the rewards interspersed between the decks are probabilistic punishments (monetary losses with different amounts). The final objective of the task is to earn as much money as possible and lose as little money as possible by choosing the cards from any deck, and participants are able to change the deck at any time. The score for this test is obtained by the difference of selected cards from decks A and B, and from decks C and D (CD–AB). Higher scores indicate better performance on the task. This means that the subject will have chosen more cards from decks C and D as they are advantageous (less penalties), while decks A and B are not advantageous (more penalties).

### 2.4. Procedure

The presence of T2D was diagnosed by a physician and the information was retrieved from medical records. Obesity was defined as BMI ≥ 30 kg/m^2^, calculated using the formula BMI = weight(kg)/(height(m))^2^. AN-R and GD samples were diagnosed according to the DSM-5 criteria [62] by clinical psychologists and psychiatrists with more than 15 years of experience in the field, during a face-to-face clinical interview. Regarding the neuropsychological evaluation, it was administered by a trained psychologist in a single session. In addition, the tests were specifically selected to determine various dimensions of executive functions. Other significant information was collected during the clinical interviews, such as sex, age, and education level.

### 2.5. Statistical Analysis

Statistical analysis was performed with Stata17 for Windows [63]. Comparison between the groups was performed with analysis of covariance (ANCOVA, adjusted for sex, age, educational level, and BMI) for quantitative measures and with logistic regression (also adjusted for the same covariates) for binary measures. Since the groups were ordered according to their position within the compulsivity–impulsivity continuous dimensional spectrum, these models included polynomial contrasts to assess the presence of patterns in data adjusted by linear, quadratic, cubic, and quartic equation/functions (four polynomial trends were assessed, the maximum allowed for variables categorized in five group levels). Pairwise post hoc comparisons also explored differences between group means and proportions.

The Finner method was employed to control type I error due to multiple null hypothesis tests. This is a correction procedure based on a stepwise multiple-test method aimed to adjust *p*-values whilst controlling the familywise error rate (FWER, defined as the likelihood of achieving at least k false rejections) [64]. Controlling the *k-*FWER implies fixing a number of *k*−1 of tolerated erroneous rejections, and then combining the unadjusted *p*-values to obtain a single testing for the group of null hypothesis tests at α-level.

## 3. Results

### 3.1. Descriptive for the Sample

Table 1 displays the distribution of the patients’ sex, education levels, age, and BMI, as well as the comparison between the groups. The AN-R group included a high proportion of women and patients with secondary or university education levels, the youngest mean age, and the lowest BMI. The OB-DM group was also characterized by a high proportion of women, patients with primary or secondary study levels, and the highest BMI. OB + DM also included mostly women, the highest proportion of participants with primary education levels, the oldest mean age, and the highest mean BMI. The GD group included mostly men and a high proportion of patients with primary education levels.

### 3.2. Comparison of Impulsivity and Compulsivity Measures

Table 2 contains the ANCOVA results with a comparison between the means registered in the impulsivity and compulsivity measures, adjusted for the covariates of sex, age, educational level, and BMI (see Figure 1 for the performance line graph for the adjusted mean scores). These results provide evidence for differences between the groups. As expected, the HC achieved the highest performance in the neuropsychological measures (highest means in the IGT and lowest means in the WCST errors and perseverative errors), the lowest mean in harm avoidance, and the lowest psychological distress. AN-R showed the worst performance in the IGT task (the scores were similar to those obtained among GD patients), the lowest mean in the novelty seeking trait, and the highest mean in harm avoidance (for this personality trait, the mean score was quite similar to OB-DM). Regarding the IGT total raw score, patients with OB + DM achieved worse performance compared to the HC group, whereas no differences were found between the participants in the OB-DM group and the HC group. The OB-DM condition reported the worst performance in the WCST task and the highest mean in the harm avoidance scale. The GD patients also achieved poor performance in the neuropsychological task, and the highest mean in the novelty seeking dimension. Figure 2 includes the line chart showing the performance learning curve in the IGT task. HC obtained the best performance, followed by OB-DM and OB + DM, while AN-R and GD achieved the worst results.

Regarding polynomial contrasts, most measures did not adjust to a linear trend, while other quadratic–cubic–quartic functions achieved statistical significance (Table 2). These results indicated that other patterns in data with many fluctuations are more likely to appear than simply increasing or decreasing means within the impulsivity–compulsivity continuous spectrum.

### 3.3. Comparison of Psychological State

Appendix A shows the comparison of the psychopathology state between the groups, according to the SCL-90R scales (see Appendix A for the T-standardized mean scores). A healthier psychology status was registered for the participants within the HC group, followed by the OB + DM patients. On the other hand, the GD and AN-R conditions registered the worse psychology state.

## 4. Discussion

In the present study, we sought to investigate cognitive and personality traits associated with impulsivity and compulsivity in individuals with obesity in the presence or absence of T2D. Additional groups included in the study were healthy, normal-weight individuals, highly impulsive patients (patients with GD), and underweight, highly compulsive patients (patients with AN-R). Individuals with obesity and T2D showed highly impulsive decision making, whereas the other measure of impulsivity, novelty seeking, was not associated with obesity with T2D, nor with obesity only. For the compulsive pole, individuals with only obesity presented poor cognitive flexibility and high harm avoidance, although these dimensions were not associated with obesity plus T2D.

Impulsive decision making (e.g., choice impulsivity) is characterized by the preference for high immediate reward, despite higher future losses, in terms of both physical and psychological outcomes [17]. Poor decision making, shown by a lower IGT total score, was observed in individuals with obesity in the presence of T2D, when compared with the HC group. This was similar to what was observed in GD and AN-R compared to the HC. By contrast, the IGT total score in individuals with obesity in the absence of T2D did not differ from that of the HC group. Our findings are consistent with a previous study [55], which showed more disadvantageous decisions in the IGT total score in individuals with obesity plus T2D than in the HC. A potential explanation for the relation between obesity plus T2D and cognitive components of impulsivity could be, to some extent, the deficiencies in central insulin signaling, which are thought to impact the brain’s dopaminergic (DA) systems [45,46,47,48,49]. Given the central role of DA in cognitive functions related to impulsivity [65,66,67], it is possible that the presence of T2D and the related alterations in insulin signaling in the brain impact these cognitive dimensions of impulsivity [68,69].

Regarding personality traits related to impulsivity, novelty seeking reflects the tendency to seek out new stimuli and experiences, to be easily bored, and be inclined to avoid monotony [70]. The group of patients with GD were the only group that showed high novelty seeking, whereas individuals with obesity in the presence or absence of T2D were not characterized by high novelty seeking when compared with the HC. Although some studies in the general population reported a positive relation between novelty seeking and BMI [70], this was not found in clinical populations of individuals with obesity, in which novelty seeking was not related to BMI [71] or to successful weight loss [72]. Moreover, it has been suggested that higher novelty seeking is more frequently associated with the presence of eating disorders (e.g., binge eating disorder and night eating disorder) [73], rather than obesity. Impulsive personality traits more strictly linked to decision making, such as urgency [74] and a lack of premeditation [75], may be expected to be more pronounced in individuals with obesity in the presence of T2D, although no studies are available to date.

For the compulsive spectrum, cognitive flexibility refers to the ability to flexibly adapt one’s behavior to a changeable environment [76]. We found poor flexibility in individuals with obesity without T2D, compared to the other groups. This is consistent with previous findings, in which deficits in cognitive flexibility have been observed in people with overweight and obesity [34,35]. Difficulties in shifting current behavior in response to different requirements could negatively impact eating behaviors, and this cognitive rigidity could help to maintain unhealthy eating habits and, thus, relate to high body weight [77].

Concerning personality traits, harm avoidance is defined as the tendency to be motivated by a desire to avoid aversive experiences, which is strictly related to compulsive attitudes. We observed higher levels of harm avoidance in the group of individuals with obesity without T2D. This is in line with previous studies in clinical samples, which showed a positive association between harm avoidance and obesity [40,41,42]. Higher harm avoidance scores have been particularly reported in patients with grade 3 obesity compared with grade 2 and 1 obesity [42]. Nevertheless, higher psychological distress was present in individuals with obesity without T2D compared to the individuals with obesity plus T2D, which could contribute to more rigid behavior and cognition.

Limits and Strengths

The present study was limited by the absence of some important variables, such as the duration of diabetes and the diabetes medication, which could have interfered with the results. Therefore, these findings should be interpreted with caution, and further studies, taking medication and illness duration into account, will need to be undertaken. Furthermore, considering the complex nature of impulsivity and compulsivity, a broader assessment of the other domains of impulsivity/compulsivity would be informative, to better characterize obesity in the presence or absence of T2D.

Despite these limitations, one of the strengths of the study is the inclusion of clinical comparison groups that are representative of impulsivity and compulsivity, such as GD and AN-R. This facilitates the placement of obesity groups along the impulsive–compulsive spectrum. An additional strength is the use of both neurocognitive and personality measures, enabling a more comprehensive assessment.

## 5. Conclusions

Taken together, the results of this study suggest that the individuals with obesity in the absence of T2D were more rigid in their behavior and showed more compulsive personality traits than those with obesity plus T2D. On the other pole of impulsivity, we found that individuals with obesity in the presence of T2D were more impulsive in their decisions compared to healthy, normal-weight controls; this would allocate them to the impulsive pole of the impulsive–compulsive spectrum. If so, the tendency to make impulsive choices may be expected to negatively impact self-control and diabetes management. However, due to the lack of information about diabetic medication and diabetes duration, which affect insulin signaling in the brain, these findings should be interpreted with caution. Further studies, controlling these variables, are needed to confirm the present findings.

Despite its exploratory nature, this study offers preliminary insights into the personality traits associated with the compulsivity–impulsivity spectrum in individuals with obesity in the presence or absence of T2D. For the health care provider, identifying and understanding the presence of personality traits that could act as a barrier to treatment adherence may improve the success rates of diabetes management and obesity weight loss treatments.

## Figures and Tables

**Figure 1 nutrients-13-04426-f001:**
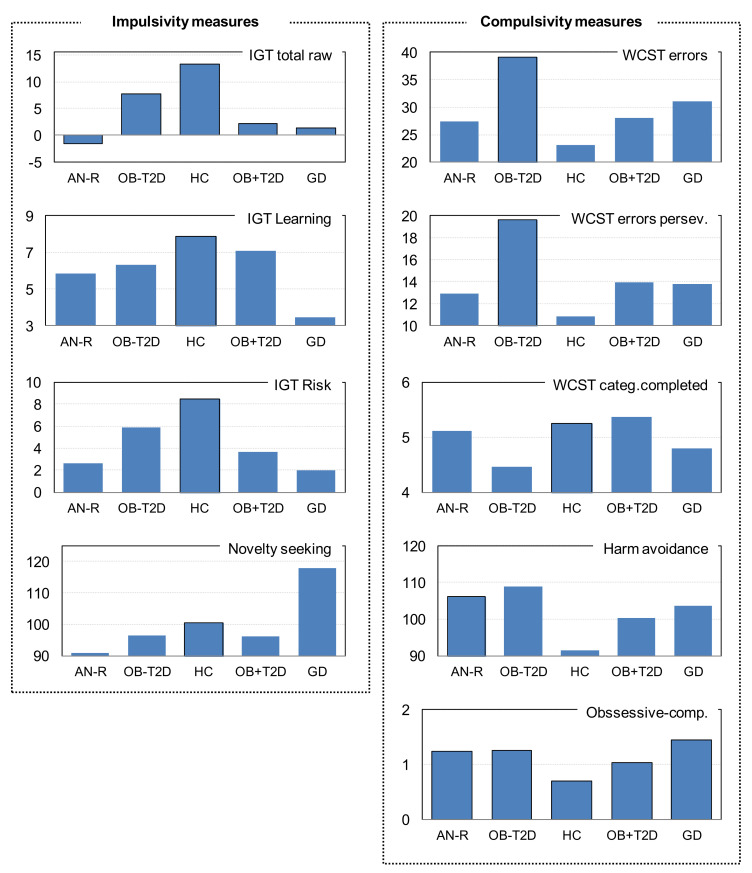
Bar charts with the impulsivity–compulsivity measures in the study (dimensional scores).

**Figure 2 nutrients-13-04426-f002:**
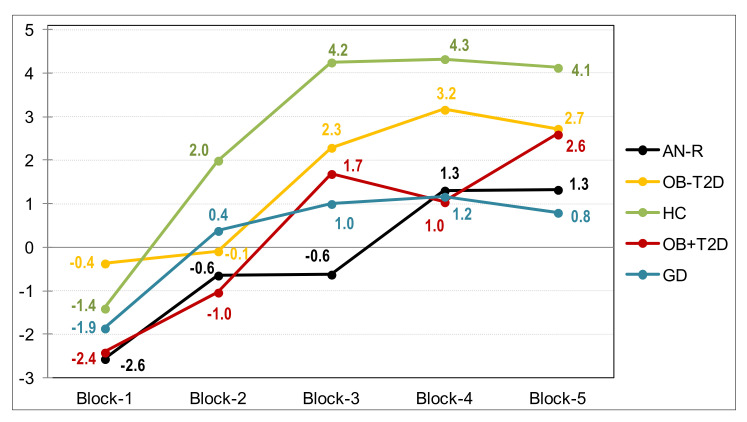
Performance learning curve (IGT task). Note. AN-R: anorexia nervosa restrictive. OB − T2D: obesity without T2D. HC: healthy control. OB + T2D: obesity with T2D. GD: gambling disorder. *Y*-axis represents the means adjusted by sex, age, education, and BMI.

**Table 1 nutrients-13-04426-t001:** Descriptive for the sample.

	AN-R		OB − T2D		HC		OB + T2D		GD		
	*N* = 107		*N* = 115		*N* = 171		*N* = 67		*N* = 121		
	*n*	*%*	*n*	*%*	*n*	*%*	*n*	*%*	*n*	*%*	*p*
Sex											
Women	97	90.7%	107	93.0%	144	84.2%	48	71.6%	19	15.7%	**<0.001 ***
Men	10	9.3%	8	7.0%	27	15.8%	19	28.4%	102	84.3%	
Education Primary	29	27.1%	48	41.7%	16	9.4%	48	71.6%	72	59.5%	**<0.001 ***
Secondary	48	44.9%	53	46.1%	104	60.8%	17	25.4%	32	26.4%	
University	30	28.0%	14	12.2%	51	29.8%	2	3.0%	17	14.0%	
	Mean	SD	Mean	SD	Mean	SD	Mean	SD	Mean	SD	*p*
Age (years)	25.31	8.30	41.39	11.87	29.71	13.28	54.61	11.33	38.30	13.56	**<0.001 ***
BMI (kg/m^2^)	16.36	2.17	44.60	6.70	22.34	3.14	41.96	8.59	26.38	5.90	**<0.001 ***

Note. AN-R: anorexia nervosa restrictive. OB − T2D: obesity without T2D. HC: healthy control. OB + T2D: obesity without T2D. GD: gambling disorder. SD: standard deviation. * Bold: significant comparison.

**Table 2 nutrients-13-04426-t002:** Comparison between the groups in impulsivity and compulsivity measurements. ANCOVA adjusted by sex, age, education and BMI.

	AN-R		OB − T2D		HC		OB + T2D		GD		Polynomial Contrasts			
	*N* = 107		*N* = 115		*N* = 171		*N* = 67		*N* = 121		Trends (*p*-Value)			
*Impulsivity*	*Mean*	*SD*	*Mean*	*SD*	*Mean*	*SD*	*Mean*	*SD*	*Mean*	*SD*	*O1*	*O2*	*O3*	*O4*
IGT Total raw	−1.62	20.33	7.71	22.12	13.24	28.52	2.10	20.31	1.41	24.58	0.964	**0.004 ***	0.106	0.308
IGT Learning	5.84	12.94	6.35	14.65	7.87	16.84	7.08	14.38	3.44	14.38	0.511	0.177	0.477	0.909
IGT Risk index	2.63	13.83	5.89	13.42	8.46	17.02	3.63	13.28	1.96	15.22	0.561	**0.029 ***	0.481	0.478
TCI-R Novelty seeking	90.94	12.56	96.53	13.21	100.33	11.45	96.15	12.11	117.82	9.29	**0.001 ***	**0.001 ***	**0.001 ***	**0.035 ***
*Compulsivity*	*Mean*	*SD*	*Mean*	*SD*	*Mean*	*SD*	*Mean*	*SD*	*Mean*	*SD*	*O1*	*O2*	*O3*	*O4*
WCST Errors	27.40	16.71	39.19	25.96	23.06	16.04	27.98	24.61	31.04	24.66	0.636	0.733	**0.001 ***	**0.027 ***
WCST Errors perseve.	12.94	7.71	19.67	14.61	10.84	7.24	13.96	13.21	13.80	10.88	0.348	0.735	**0.001 ***	**0.011**
WCST Categ.compl.	5.11	1.20	4.45	2.22	5.25	1.21	5.37	2.12	4.80	1.97	0.650	0.549	**0.001 ***	0.416
TCI-R Harm avoid.	106.02	18.63	108.88	16.87	91.38	16.21	100.21	14.07	103.51	18.77	**0.046 ***	**0.002 ***	**0.015 ***	**0.004 ***
SCL-90R Obs.-comp.	1.24	0.91	1.26	0.73	0.70	0.56	1.04	0.67	1.45	0.87	0.505	**0.001 ***	**0.016 ***	0.055
Pairwise	AN-R/		AN-R/		AN-R/		AN-R/	OB − T2D/	OB − T2D/	OB − T2D/	HC/	HC/	OB + T2D/	
**comparisons**	OB − T2D		HC		OB + T2D		GD	HC	OB + T2D	GD	OB + T2D	GD	GD	η^2^
*Impulsivity*														
IGT Total raw	0.142		**<0.001 ***		0.572		0.493	0.291	0.159	0.224	**0.046 ***	**0.002 ***	0.895	0.048
IGT Learning	0.897		0.307		0.760		0.381	0.638	0.764	0.365	0.821	0.056	0.259	0.008
IGT Risk index	0.407		**0.004 ***		0.806		0.807	0.429	0.361	0.222	0.164	**0.005 ***	0.605	0.024
TCI-R Novelty seeking	0.068		**<0.001 ***		0.102		**<0.001 ***	0.133	0.843	**<0.001 ***	0.121	**<0.001 ***	**<0.001 ***	**0.282 ^†^**
*Compulsivity*														
WCST Errors	**0.025 ***		0.104		0.915		0.321	**<0.001 ***	**0.001 ***	0.059	0.288	**0.010 ***	0.479	0.043
WCST Errors perseve.	**0.013 ***		0.125		0.715		0.647	**<0.001 ***	**0.001 ***	0.008 *	0.189	0.063	0.943	0.042
WCST Categ.compl.	0.126		0.496		0.555		0.306	**0.023 ***	**0.001 ***	0.318	0.761	0.072	0.107	0.029
TCI-R Harm avoid.	0.512		**<0.001 ***		0.199		0.408	**<0.001 ***	**0.002 ***	0.132	**0.022 ***	**<0.001 ***	0.358	**0.124 ^†^**
SCL-90R Obs.-comp.	0.937		**<0.001 ***		0.314		0.119	**<0.001 ***	0.073	0.218	**0.046 ***	**<0.001 ***	**0.010 ***	**0.109 ^†^**

Note. AN-R: anorexia nervosa restrictive. HC: healthy control. GD: gambling disorder. OB − T2D: obesity without T2D. OB + T2D: obesity with T2D. IGT: Iowa gambling test. SCL-90: Symptom Checklist—90 Items—Revised. TCI-R: Temperament and Character Inventory–Revised. WCST: Wisconsin card sorting test. SD: standard deviation. O1: order 1, linear. O2: order 2, quadratic. O3: order 3, cubic. O4: order 4, quartic. η^2^: partial eta squared. * Bold: significant parameter. **^†^** Bold: effect size within the ranges moderate/medium to large/high. Note. AN-R: anorexia nervosa restrictive. OB − T2D: obesity without T2D. HC: healthy control. OB + T2D: obesity with T2D. GD: gambling disorder. IGT: Iowa gambling test. WCST: Wisconsin card sorting test. *Y*-axis represents the means adjusted by sex, age, education, and BMI.

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
