# Peer review of "Transdiagnostic Perspective of Impulsivity and Compulsivity in Obesity: From Cognitive Profile to Self-Reported Dimensions in Clinical Samples with and without Diabetes"

_nutrients, 2021, doi:10.3390/nu13124426_

Round 1

Reviewer 1 Report

Introduction is missing the primary cause and classification of obesity and its litany of lethal comorbidities, Maillard Abuse Disorder, and other Food Misuse Disorders (Cocores JA, Gold MS. The Salted Food Addiction Hypothesis may explain overeating and the obesity epidemic, 2009). Browned or Maillard-coated foods such as chips stimulate oral mesolimbic opiate/dopamine receptors producing dose related progression and euphoria, along the same neuropsychiatric pathways and destinations as heroin and cocaine. The Maillard-induced high is followed by Maillard-induced opiate/dopamine withdrawal, perceived as mindful images of acquiring a Maillard-coated snack, beverage, or meal linked to false perceptions of increased appetite or hunger.

‘Understanding the neurobehavioral mechanisms underpinning obesity is crucial to develop effective specific treatments’ only in the presence of a discussion surrounding Maillard abuse disorder (searched as Maillard, advanced glycation, advanced lipoxidation end-products) and its enormously well documented progression to redox imbalance, overweight, obesity, morbid obesity, immune suppression, autoimmune disease, increased risk for COVID-19 sepsis and non-SARS lung disease, hypertension, heart disease, stroke, cancer, ADD/accidents/trauma, suicide, and other psychiatric anomalies including anorexia and bulimia, T2 diabetes, Alzheimers, and renal disease.

Both ‘constructs that have been suggested to play a role in excessive food intake and weight gain are impulsivity and compulsivity’ without discussing the differences and similarities or lack of differences and similarities between impulsivity and compulsivity in illness-free people and OCD people, often borderline psychotic and body dysmorphic anorexia nervosa, gambling disorder, and rapidly cycling euphoria/withdrawal and redox imbalanced Maillard misusers.

Author Response

Point 1. Introduction is missing the primary cause and classification of obesity and its litany of lethal comorbidities, Maillard Abuse Disorder, and other Food Misuse Disorders (Cocores JA, Gold MS. The Salted Food Addiction Hypothesis may explain overeating and the obesity epidemic, 2009). Browned or Maillard-coated foods such as chips stimulate oral mesolimbic opiate/dopamine receptors producing dose related progression and euphoria, along the same neuropsychiatric pathways and destinations as heroin and cocaine. The Maillard-induced high is followed by Maillard-induced opiate/dopamine withdrawal, perceived as mindful images of acquiring a Maillard-coated snack, beverage, or meal linked to false perceptions of increased appetite or hunger. Understanding the neurobehavioral mechanisms underpinning obesity is crucial to develop effective specific treatments’ only in the presence of a discussion surrounding Maillard abuse disorder (searched as Maillard, advanced glycation, advanced lipoxidation end-products) and its enormously well documented progression to redox imbalance, overweight, obesity, morbid obesity, immune suppression, autoimmune disease, increased risk for COVID-19 sepsis and non-SARS lung disease, hypertension, heart disease, stroke, cancer, ADD/accidents/trauma, suicide, and other psychiatric anomalies including anorexia and bulimia, T2 diabetes, Alzheimers, and renal disease. 

RESPONSE: We thank the reviewer for his/her important suggestions. Accordingly, we have included the primary causes and classification of obesity in the introduction. As suggested, we also mentioned possible factors implicated in overeating and obesity, including reward-related mechanisms and opiate/dopamine alterations induced by certain foods (lines: 107-113). 

Point 3. Both ‘constructs that have been suggested to play a role in excessive food intake and weight gain are impulsivity and compulsivity’ without discussing the differences and similarities or lack of differences and similarities between impulsivity and compulsivity in illness-free people and OCD people, often borderline psychotic and body dysmorphic anorexia nervosa, gambling disorder, and rapidly cycling euphoria/withdrawal and redox imbalanced Maillard misusers. 

RESPONSE: We agreed with the reviewer about the importance of mentioning other disorders related to impulsivity and compulsivity. Accordingly, mental disorders mainly characterized by impulsive or compulsive features have been included in the introduction (lines: 140-146).

Reviewer 2 Report

In the introduction, please elaborate on what a dimensional model is,  and how they are used in clinical practice or research. You start discussing them but without any real background which may lose some readers.

Lines 154-156 do  you have evidence for these assertions? Does the literature make this association already, or are you postulating it in your introduction based on previous papers ergot, it's your research question.

If the latter, this would be considered generation of results and requires incorporating into your methods, and with an explanation as to why you have come to this conclusion.

If the former, the reword your paragraph and cite accordingly. 

Do you have information on medications for those with obesity and T2D? This is important and a potentially major confounding factor.

Lack of information on medication and duration of illness, especially when you imply insulin-signalling is involved in various aspects of cognitive control means no meaningful conclusions can be drawn. Physiological manipulation by or endocrine parameters (diet and/or drugs), which is very likely to be the case very much casts doubt here. 

You aren't able to establish a cause and effect relationship here and your conclusions are far too strong given the quality of the data.

Major revision is needed to to reframe your outcomes and what they could mean - I simply don' think you can come to conclusions you have, based on the data you have.

Author Response

Point 1. In the introduction, please elaborate on what a dimensional model is, and how they are used in clinical practice or research. You start discussing them but without any real background which may lose some readers. 

RESPONSE: Thank you for highlighting this important point. A more detailed description of what a dimensional model is and its clinical/research implications has been added to the introduction (lines 131-137).

Point 2. Lines 154-156 do you have evidence for these assertions? Does the literature make this association already, or are you postulating it in your introduction based on previous papers ergot, it's your research question. If the latter, this would be considered generation of results and requires incorporating into your methods, and with an explanation as to why you have come to this conclusion. If the former, the reword your paragraph and cite accordingly. 

RESPONSE: Thanks for your important suggestion. Concerning the main hypothesis: given that compulsivity features have been described in previous literature in individuals with obesity we expected to confirm this. Secondly, our hypothesis to find more pronounced impulsivity in obesity with T2D was based on some previous studies showing more impulsive choices when comparing individuals with T2D and healthy population. For the seeking of clarity, we restructured the section of research hypothesis, as suggested (lines: 175-180).

Point 3. Do you have information on medications for those with obesity and T2D? This is important and a potentially major confounding factor.  Lack of information on medication and duration of illness, especially when you imply insulin-signalling is involved in various aspects of cognitive control means no meaningful conclusions can be drawn. Physiological manipulation by or endocrine parameters (diet and/or drugs), which is very likely to be the case very much casts doubt here. 

RESPONSE: We completely agreed with your comment. The lack of information about medication and illness duration is an important limitation of the present work, that we described in the limitation session of the discussion. In order to remark on this crucial aspect, we also specified in the final conclusion, the need for future studies taking into account these variables (lines: 404-413).

Point 4. You aren't able to establish a cause and effect relationship here and your conclusions are far too strong given the quality of the data. Major revision is needed to to reframe your outcomes and what they could mean - I simply don' think you can come to conclusions you have, based on the data you have.

RESPONSE: We totally agree with you about the important limitations of the study. Despite this, these results should be considered preliminary in their nature, since is the first study investigating impulsivity and compulsivity dimensions in groups of obese individuals in presence or absence of T2D, also considering other clinical groups as a reference. Accordingly, results are discussed as possible explanations which the data suggest, carefully considering the limitations (lines: 404-413).

Round 2

Reviewer 1 Report

This paper appears ready for publication